# Potential impact and cost-effectiveness of long-acting injectable lenacapavir plus cabotegravir as HIV treatment in Africa

Andrew Phillips [1] ✉, Jennifer Smith[1], Loveleen Bansi-Matharu[1], Kenly Sikwese[2], Cissy Kityo[3], Charles Flexner[4], Marco Vitoria[5], Nathan Ford[5], Meg Doherty[5], Zack Panos[6], David Ripin[7], Matthew Hickey [8], Diane Havlir[8], Monica Gandhi [8], Michael Reid[8,9] & Paul Revill[10]

Although viral suppression is attained for most adults living with diagnosed HIV in East, Central, Southern and West Africa (ECSWA), challenges remain with sustained adherence to daily oral pill taking for some in the population. Here, we evaluate the potential effectiveness and cost-effectiveness of introduction of a new combination of long-acting injectable drugs of lenacapavir + cabotegravir to increase levels of sustained viral suppression. We find there is potential for a significant impact on HIV deaths and disability adjusted life years, including due to a decrease in mother to child transmission. If lenacapavir + cabotegravir can be sourced at a cost of around $ 80 per year or less, our analysis suggests there is potential for a policy to introduce it to be cost-effective in settings in ECSWA. Recognising the limitations of a modelling study, we suggest that implementation studies be conducted to confirm the viability of these approaches.

HIV incidence in Africa has declined in recent years but remains substantial, particularly in southern Africa[1]. A major reason for the decline is the success of providing oral antiretroviral drugs for people with HIV (PWH), as treatment as prevention is a powerful strategy to reduce HIV incidence. In people with drug sensitive virus who are adherent to daily pill taking these drugs lead to viral suppression, eliminating onward HIV sexual transmission risk[2]. However, for various individual, contextual, and health service-related reasons, not everybody is able to adhere to daily pill taking. In such cases, a regimen consisting of two long-acting injectable drugs may be a suitable alternative[3–19]. This has the advantage of removing the need for daily pill taking, although it may involve more frequent clinic visits. While currently long-acting treatment is not available in most African settings, a dual long-acting drug regimen of the integrase inhibitor cabotegravir plus the non-nucleoside reverse transcriptase inhibitor(NNRTI) rilpivirine has been

approved for use in some high income countries and has recently been shown to be non-inferior to oral therapy at 48 weeks among virologically suppressed individuals in a trial in Uganda, Kenya and South Africa[10–21]. Despite this trial conducted in Africa, cost and accessibility of long-acting cabotegravir/rilpivirine has so far limited its availability.

In order to further consider the possibility of future introduction of long-acting treatment in African settings, therefore, it is important to assess under what conditions it is likely to be cost-effective. We previously used an individual-based model of HIV in the context of East, Central Southern and West Africa (ECSWA) to model the possible impact and cost-effectiveness of cabotegravir + rilpivirine[22]. A major drawback of rilpivirine in the African context is that it is an NNRTI which has cross-resistance with the formerly widely used drug efavirenz. In addition, long-acting rilpivirine requires cold chain preservation, making it impractical for most low-income countries. Nevertheless, we

[1]Institute for Global Health, University College London, London, UK. [2]AfroCAB, Lusaka, Zambia. [3]Joint Clinical Research Centre, Kampala, Uganda. [4]Johns Hopkins University School of Medicine, Baltimore, MD, USA. [5]Department of Global HIV, Hepatitis and Sexually Transmitted Infections Programmes, WHO, Geneva, Switzerland. [6]Children's Investment Fund Foundation, London, UK. [7]Clinton Health Access Initiative, New York, NY, USA. [8]University of California San Francisco, San Francisco, CA, USA. [9]US Department of State's Bureau of Global Health Security and Diplomacy – United States President's Emergency Plan for AIDS Relief (PEPFAR), Washington DC, USA. [10]University of York, York, UK. ✉e-mail: andrew.phillips@ucl.ac.uk

found cabotegravir + rilpivirine to be potentially cost-effective at an annual cost of $120 per year if its use were focused exclusively in those people on ART with viral load level >1000 copies/mL[22]. Although cabotegravir + rilpivirine has not been studied in a randomized controlled trial (RCT) in those with viral loads that are over 1000 copies/mL, demonstration projects have evaluated the utilization of long-acting regimens in people with HIV without virologic suppression and demonstrated the ability of long acting ART to achieve and maintain virologic suppression[11–15]. Subsequently, in the U.S., the IAS-USA Guidelines Panel[23] and the DHHS Guidelines for the Use of Anti-retroviral Agents in Adults and Adolescents With HIV committee[24] have now added long-acting cabotegravir+ rilpivirine to their guidelines for those with virologic failure, adherence challenges to oral ART, and a high risk of HIV progression.

Lenacapavir is a long-acting medication in the capsid inhibitor class that has now been studied in the context of HIV prevention[25], treatment for highly treatment experienced PWH[26,27], and in a small case series (n = 34) combined with cabotegravir, where the combination has shown promise in achieving virologic suppression[28]. We here aim to explore the potential impact on viral suppression, HIV incidence, HIV-related deaths, DALYs, and cost-effectiveness of a long-acting regimen of lenacapavir + cabotegravir in the context of ECSWA. Critically, the regimen has the potential advantage that lenacapavir, which requires a 6 monthly sub-cutaneous injection, is a capsid inhibitor without cross-resistance with other drugs used as treatment. The key outcomes of the current study and their implications to potentially inform policy in HIV in ECSWA are displayed in Table 1.

## Results

### Setting-scenarios

Through sampling of parameter values (see Supplementary Model Details) at the start of each model run we create 1000 "setting-scenarios" reflecting uncertainty in assumptions and a range of characteristics similar to those seen in ECSWA (Table 2). These represent sub-settings within countries as well as countries as a whole. We show national data from PHIA surveys which are generally within the range of the settings-scenarios.

### Outcomes over 10 years

For each setting-scenario, we simulate predicted outcomes with and without a policy of introduction of lenacapavir + cabotegravir treatment from 2027 onwards. The predicted effects of the policy of introduction of lenacapavir + cabotegravir over 10 years are shown in Table 3 and Fig. 1. In Table 3 we present medians and 90% range over setting scenarios as well as mean over setting scenario with 95% confidence interval for the mean. In this text we mention only the medians and 90% range over setting scenarios. With the introduction of lenacapavir + cabotegravir, of PWH on ART, the percentage who are on lenacapavir + cabotegravir is a median 15% over setting-scenarios (90% range, 5%–37%) over these 10 years. Of PWH on lenacapavir+cabotegravir, the percentage who started due to viral non-suppression on current oral drugs is 24% (7%–55%), the percentage starting after having been out of care and brought back in due to offer of lenacapavir + cabotegravir is 40% (14%–75%), with the remainder having started when virally suppressed or ART naïve due to a strong preference expressed for an injectable regimen.

Of PWH on lenacapavir + cabotegravir, the percentage with viral load <1000 copies/mL is 96% (90%–99%). The percentage of PWH on lenacapavir + cabotegravir with a capsid inhibitor / integrase inhibitor drug resistance mutation is 0.2% (0.0%–1.4%) / 0.9% (0.2%–3.0%). Of all PWH, the percentages with viral load <1000 copies/mL are 86% (77%–92%) without lenacapavir + cabotegravir introduction and 87% (80%–93%) with lenacapavir + cabotegravir, respectively. Of PWH with diagnosed HIV, the percentage with viral load >1000 copies/mL

## Table 1 | Policy summary and key outcomes

| | |
|---|---|
| • Background | Use of antiretroviral treatment in people with HIV (PWH) to suppress viral replication has been critical for helping to reduce HIV incidence as well as deaths from HIV. Although viral suppression is attained for most adults living with diagnosed HIV in East, Central, Southern and West Africa (ECSWA), challenges remain with sustained adherence to daily oral pill taking for some in the population. Long-acting injectable treatment could offer an effective alternative in such people, with lenacapavir + cabotegravir being a possible regimen option. In order to explore this, we used an existing individual-based model of HIV in the ECSWA context to assess potential effectiveness and cost-effectiveness of a policy of introduction of lenacapavir + cabotegravir with the aim of increasing levels of sustained viral suppression in PWH. We assume that the policy would involve active offer of lenacapavir + cabotegravir in people with sustained viral load measured >1000 copies/mL (despite enhanced adherence counselling) on oral drugs, and to people living with diagnosed HIV who are not currently engaged in treatment. |
| • Main findings and limitations | Our modelling analysis suggests that there are substantial potential health benefits from introducing lenacapavir + cabotegravir long-acting treatment in ECSWA settings. Across all setting scenarios, our assumptions on uptake led to a median 15% of people on ART being on lenacapavir + cabotegravir over the first 10 years from its introduction. Given this, there was a decrease in median percentage of diagnosed PWH with viral load >1000 copies/mL from 8.3% to 6.9% over the first 10 years (a 17% reduction). This was predicted to lead to a median 19% reduction in HIV deaths and a 18% reduction in mother to child transmission of HIV. Implementation costs are uncertain, but at an average total annual cost per person of $ 140 we found that introduction of lenacapavir + cabotegravir in settings with percentage of diagnosed PWH with viral load <1000 copies/mL below 93% is likely to be cost-effective in the context of a cost-effectiveness threshold of $ 500/DALY averted. If the cost could be $ 100 per year then lenacapavir + cabotegravir introduction is likely to be cost-effective in almost all settings. If targeted at women aged 15–39 or young people aged 15–24 the introduction of lenacapavir + cabotegravir is predicted to be cost-effective even in the context of a cost-effectiveness threshold of $ 150. |
| • Policy implications | We suggest that pilot implementation studies be conducted to confirm the viability of implementation. Such studies are needed to further understand whether introduction of lenacapavir + cabotegravir has potential. It is important that there is community engagement at every stage, especially since this might help to manage and address some of the equity issues but also prepare, raise awareness and incentivize targeted PWH eligible to access long acting injectable treatment. Implementation studies might initially recruit people who are attending clinic but self-report poor adherence to oral medication and have unsuppressed viral load. As more experience is gained and if and when it becomes clear that the injections can be consistently delivered in clinics and viral suppression attained in such PWH then this would provide evidence to support roll out of offer of lenacapavir + cabotegravir in this group. Implementation studies might then move on to studies which seek out people who have dropped out of care to offer them the option of lenacapavir + cabotegravir. This would initially be in people who are unlikely to be mobile who could likely be found if they did not return to care after their first injection in order to ascertain reasons. It may be possible to explore community-based delivery of injections. Frequent viral load monitoring, ideally at point of care, and resistance testing will be important in such studies to check that the regimen is leading to sustained viral suppression and not associated with development of drug resistance mutations. While implementation studies are needed before a recommendation can be made to introduce long acting treatment in the way proposed, policy-makers should begin to consider how they might make the proposed regimen available, including consideration of whether this might be done in community settings. |

**Table 2 | Description of setting-scenarios in 2024. Based on $n = 1000$ setting-scenarios**

| Characteristic | Model (median, 90% range) | Examples of observed data[a] |
|---|---|---|
| HIV prevalence (all / women / men) age 15–49 | 9.9% (0.3% - 22.6%)<br>2.9% (4.7% - 30.5%)<br>6.6% (2.2% - 15.4%) | Zimbabwe 2020 (women/men) 15%/9%, U Rep Tanzania 2023 5%/2%, Uganda 2020 7.1%/3.8%, Lesotho 2020 28%/16%, Eswatini 2021 32%/16%, Malawi 2020 10%/6%, Namibia 2017 15%/8%, Zambia 2021 13%/6%, Cameroon 2018 3%/2%, Cote d'Ivoire 2017/18 4%/1%, Rwanda 2019 2.6%, Kenya 2018 (age 15-64) 6.6% / 3.1%, South Africa 2022 16%/9%. |
| HIV incidence (/100 person years) (all / women / men) age 15–49 | 0.37 (0.08 – 1.27)<br>0.47 (0.10 – 1.78)<br>0.28 (0.06 – 0.88) | Malawi 2016 (women/men) 0.44/0.22 2020 0.31/0.15, Zambia 2021 0.63/0.05, Zimbabwe 2020 0.67/0.23, Lesotho 2020 0.81/0.33, Eswatini 2021 1.45/0.20, Tanzania 2023 0.29/0.12, Cameroon 2017 0.40/0.08 Rwanda 2019 0.08 Uganda 2020 0.42/0.21, Kenya 2018 (age 15–64) 0.14. South Africa 0.87/0.64. |
| Percentage of HIV positive people diagnosed (all / women / men) | 92% (85% - 97%)<br>94% (87% - 98%)<br>87% (78% - 94%) | Malawi (women/men) 2020 90%/85%, Zambia 2021 90%/87%, Zimbabwe 2020 88%/84%, Namibia 2017 (age 15–64) 90%/80%, Tanzania 2017 55%/45% 2023 85%/78%, Ethiopia (age 15–64) 2018 83%/70%, Cote d'Ivoire 2017/18 (age 15–64) 43%/24%, Cameroon 2017 (age 15–64) 58%/51%, Mozambique 2021 73%/69%, Uganda 2021 84%/76%, Rwanda 2019 86%/80%, Eswatini 2021 95%/92%, Lesotho 2020 91%/88%, Kenya 2018 (age 15–64) 83%/73% South Africa 2022 92%/85%. |
| Percentage of diagnosed HIV positive people on ART (all / women / men) | 96% (89% - 98%)<br>96% (90% - 98%)<br>95% (86% - 98%) | $Lesotho (women/men) 2020 98%/96%, South Africa 2022 91%/90%, Eswatini 2021 98%/96%, Namibia 2017 97%/95% (age 15–64), Zambia 2021 98%/98%, Tanzania 2023 98%/97%, Ethiopia 2018 (age 15–64) 96%/99%, Malawi 2020 98%/97%, Uganda 2021 97%/95%, Cameroon 2017 (age 15–64) 93%/94%, Zimbabwe 2020 98%/96%, Cote d'Ivoire 2017/18 (age 15–64) 93%/71%, Mozambique 98%/94%, Rwanda 2018 98%/97%, Kenya 2018 97%/95%. |
| Of people on ART, percentage with VL < 1000 (all / women / men) | 95% (89% - 98%)<br>96% (91% - 98%)<br>93% (84% - 97%) | $Zambia (women/men) 2021 96%/97%, Malawi 2020 97%/97%, Zimbabwe 2020 91%/89%, Namibia 2017 92%/90%, Tanzania 2023 95%/93%, Ethiopia 2018 (age 16–64) 86%/91%, Cote d'Ivoire 2017/18 (age 15–64) 78%/65%, Cameroon 2017 80%/81%, Mozambique 2021 90%/88%, Uganda 2021 93%/91%, Rwanda 2018 92%/85%, Eswatini 2021 96%/98%, Lesotho 2020 92%/90%, Kenya 2018 90%/91% South Africa 2022 94%/94%. |
| Of people with diagnosed HIV, percentage with VL < 1000 (all / women / men) | 89% (81% - 94%)<br>91% (84% - 96%)<br>87% (77% - 93%) | $ Zimbabwe 2020 89%/85%, Zambia (women/men) 2021 94%/95%, Malawi 2020 95%/94%, Namibia 2017 89%/85%, Tanzania 2023 93%/90%, Ethiopia 2018 (age 16–64) 82%/90%, Cote d'Ivoire 2017/18 (age 15-64) 75%/46%, Cameroon 2017 73%/75%, Mozambique 2021 88%/82%, Uganda 2021 90%/86%, Rwanda 2018 90%/82%, Eswatini 2021 94%/94%, Lesotho 2020 90%/86%, Kenya 2018 87%/86% South Africa 2022 85%/84%. |
| Percentage of all HIV positive people with VL < 1000 copies/mL (all / women / men) | 81% (72% - 89%)<br>85% (77% - 91%)<br>74% (61% - 84%) | Zambia 2021 86%, Malawi 2020 87%, Zimbabwe 2020 76%, Eswatini 2021 87%, Lesotho 2020 81%, Tanzania 2023 78%, Uganda 2020 75%, Namibia 2017 (age 15–64) 77%, Ethiopia 2018 (age 15–64) 70%, Cote d'Ivoire 2017/18 (age 15–64) 40%, Cameroon 2017 (age 15–64) 47% Rwanda 2019 76%, Kenya 2018 72%. |
| Prevalence of HIV viral load >1000 copies/mL amongst all adults | 2.1% (0.7% - 5.4%) | Zambia 2021 1.4%, Namibia 2017 2.8% (age 15–64), Malawi 2020 1.2%, Zimbabwe 2020 3.1% (age 15 +), Cote d'Ivoire 2018 1.7% (age 15-64), Eswatini 2021 3.2%, Lesotho 2020 4.3%. |

[a] all observed data from PHIA surveys (Population Health Impact Assessments) https://phia.icap.columbia.edu/[56], file:///C:/Users/w3sth/UCL%20Dropbox/Andrew%20Phillips/PC/Downloads/ SABSSMVI-SUMMARY-SHEET-2023.pdf. Note that we show national data from countries, but setting scenarios are conceived of as reflecting also sub-settings within countries, not only countries as a whole. $ adjusted for having a detectable antiretroviral in blood. Setting-scenarios were restricted to those with HIV prevalence <35% in women, <25% in men, HIV incidence <1.5 in men <2.5 in women, percentage of PWH diagnosed 75% for women and 70% for men, percentage on ART of those with diagnosed HIV >80% in women and >73% in men, and percentage of those on ART with viral lad <1000 cps/mL >70%, and with higher ART coverage, HIV incidence and HIV prevalence in women compared with men. All outputs refer to adults age 15+ unless otherwise stated.

decreases from 8.3% (4.6%–15.7%) without lenacapavir + cabotegravir introduction to 6.9% (3.9%–13.0%) with lenacapavir + cabotegravir introduction.

The total number of HIV related deaths per year amongst all PWH per 1 million adults (i.e. in the context of a setting with 1 million adults aged 15+ in 2024) is 1030 (340 – 2620) with no introduction of lenacapavir + cabotegravir and 830 (290–2010) with its introduction (percent reduction 19% (0%–35%)). The prevalence of HIV viral load >1000 copies/mL amongst all adults is predicted to be 1.3% (0.4% –4.0%) without lenacapavir + cabotegravir introduction and 1.2% (0.4%–3.5%) with, a relative prevalence of 0.90 (0.74–1.10). HIV incidence in women aged 15–49 (per 100 person years) is predicted to be 0.28 (0.06–1.28) over the 10 year period compared with 0.27 (0.05–1.16) with lenacapavir + cabotegravir introduction (relative incidence over setting-scenarios 0.95 (0.67–1.30)), while in men the corresponding values are 0.18 (0.04–0.66), 0.17 (0.03–0.59) and 0.94 (0.60–1.45). Finally, the percentage of births in women with HIV in which the child acquires HIV (either at birth or through breastfeeding) is predicted to be 5.1% (2.1% - 10.1%) without introduction of lenacapavir + cabotegravir and 4.0% (1.7%–8.3%) with its introduction (percent reduction 0.82 (0.54–1.20))

We fitted a series of logistic regression models across setting-scenarios to evaluate characteristics of setting-scenarios in 2024 predicting a >15% decline in deaths with lenacapavir + cabotegravir introduction (Supplementary Table S30). The strongest predictors were the percentage of diagnosed PWH on ART and the percentage of PWH on ART with viral load <1000 copies/mL.

**Budget impact and cost-effectiveness**

The annual cost of providing HIV programmes in the 3 years from 2027, without introduction of lenacapavir + cabotegravir (in the context of a setting with a population of 1 million adults) is $19.94 million per year. The budget impact of introduction of lenacapavir + cabotegravir in 2027 at the costs assumed would be an increase of $ 0.69 million per year to $ 20.63 million, an increase of 3.5%.

Over a 50 year time horizon, discounted annual costs with introduction of lenacapavir + cabotegravir are predicted to be $ 0.67 million higher in a population of 1 million adults. This is due to the higher cost of antiretroviral drugs with lenacapavir + cabotegravir introduction ($ 3.71 m vs $ 3.35 m) and also driven by increased clinic visit costs ($ 1.73 m vs $ 1.23 m), which remain especially uncertain and might fall with innovations in delivery (Table 4) Over this time horizon (Table 5)

**Table 3 | Predicted effects of introduction of lenacapavir + cabotegravir over 10 years**

| Model output | No lenacapavir + cabotegravir introduction | Lenacapavir-cabotegravir introduction |
|---|---|---|
| Of PWH on ART, percentage who are on lenacapavir+cabotegravir<br>Age 15+<br>Ages 15–24<br>Women age 15+<br>Men age 15+ | ---<br>---<br>---<br>--- | 15% (5% – 37%) 17% (17% - 18%)<br>20% (7% - 40%) 21% (20% - 22%)<br>14% (5% - 35%) 17% (16% - 17%)<br>15% (5% - 36%) 17% (17% - 18%) |
| Of PWH on lenacapavir+cabotegravir, percentage who started (a) when on ART with measured viral non-suppression (b) when off ART (c) when already virally suppressed | ---<br>---<br>--- | 24% (7% – 55%) 27% (26% - 28%)<br>40% (14% – 75%) 42% (40% - 43%)<br>31% (0% - 75%) 32% (31% - 34%) |
| Of PWH on lenacapavir + cabotegravir, percentage with viral load <1000 copies/mL | --- | 96% (90% – 99%) 95% (95% - 96%) |
| Of PWH who have ever taken lenacapavir + cabotegravir:<br>percentage currently still on lenacapavir + cabotegravir<br>percentage virologically failed lenacapavir + cabotegravir | ---<br>--- | 88% (43% - 98%) 81% (79% - 82%)<br>0% (0% - 0%) 0% (0% - 0%) |
| Percentage of PWH with a capsid inhibitor/integrase inhibitor drug resistance mutation | 0.0% (0.0% - 0.0%) 0.0% (0.0% -0.0%)<br>0.9% (0.2% - 3.0%) 1.1% (1.1% - 1.2%) | 0.2% (0.0% – 1.4%) 0.4% (0.4% - 0.4%)<br>0.9% (0.2% – 3.0%) 1.1% (1.1% - 1.2%) |
| Of PWH on ART, percentage with viral load <1000 copies/mL | 97% (92% - 98%) 96% (96% - 96%) | 98% (94% - 99%) 97% (97% - 97%) |
| Of people with diagnosed HIV, percentage with viral load >1000 copies/mL<br>Difference | 8.3% (4.6% - 15.7%)<br>9.0% (8.8% - 9.2%) | 6.9% (3.9% - 13.0%)<br>7.5% (7.4% - 7.7%)<br>-1.3% (-3.9% - +0.5%)<br>-1.5% (-1.4% - -1.6%) |
| Of all PWH, percentage with viral load <1000 copies/mL | 86% (77% – 92%) 85% (85% - 86%) | 87% (80% – 93%) 87% (87% - 87%) |
| Prevalence of HIV viral load >1000 copies/mL amongst all adults<br>Relative prevalence | 1.3% (0.4% - 4.0%) 1.7% (1.6% - 1.8%) | 1.2% (0.4% - 3.5%) 1.5% (1.4% - 1.6%)<br>0.90 (0.74 – 1.10)<br>0.90 (0.89 – 0.91) |
| Number of HIV related deaths per year ^<br>Deaths averted<br>Percent reduction | 1030 (340 – 2620)<br>1210 (1160 – 1250) | 830 (290 – 2010)<br>970 (9,300 – 1010)<br>180 (0 – 680) 240 (230 – 250)<br>19% (0% – 35%) 18% (18% - 19%) |
| HIV incidence in women (age 15-49) (per 100 person years)<br>Relative rate<br>HIV incidence in men (age 15-49) (per 100 person years)<br>Relative rate | 0.28 (0.06 – 1.28) 0.41 (0.39 – 0.44)<br>---<br>0.18 (0.04 – 0.66)<br>0.25 (0.23 – 0.26)<br>--- | 0.27 (0.05–1.16) 0.39 (0.37 – 0.41)<br>0.95 (0.67 – 1.30)<br>0.97 (0.95 – 0.98)<br>0.17 (0.03 – 0.59)<br>0.23 (0.22 – 0.24)<br>0.94 (0.60 – 1.45)<br>0.97 (0.95 – 0.98) |
| Percentage of children of women with HIV for which the child is infected at birth or through breastfeeding ^<br>Number of newly infected children per year ^<br>Relative risk | 5.1% (2.1% – 10.1%)<br>5.5% (5.3% - 5.6%)<br>370 (65 – 1940)<br>600 (560 – 640) | 4.0% (1.7% – 8.3%)<br>4.4% (4.3% - 4.6%)<br>300 (50 – 1660)<br>480 (450 – 520)<br>0.82 (0.54 – 1.20)<br>0.84 (0.83 – 0.86) |

Median 90% range over setting-scenarios and *mean across setting scenarios (95% confidence interval)* are shown.

^ In the context of a setting with population size of 1 million adults aged 15+ in 2024; for a setting with x.x million adults the number can be multiplied by x.x. All outputs refer to adults age 15+ unless otherwise stated.

the mean number of HIV-related deaths averted per year with introduction of lenacapavir + cabotegravir is predicted to be 195, with 2400 DALYs averted with discounting.

The incremental cost-effectiveness ratio for introducing lenacapavir + cabotegravir is $280 per DALY averted. Using a cost-effectiveness threshold of $500, the number of net DALYs averted per year over 50 years is 1060. Considering setting-scenarios individually, the policy of lenacapavir + cabotegravir introduction is predicted to lead to lower cost in 16%, lower DALYs in 82% and lower net DALYs (i.e. is cost-effective) in 61% of setting scenarios. In a sensitivity analysis we calculated DALYs including years of life lost beyond the end of the time horizon and the ICER was little changed at $272. In a further sensitivity analysis we assessed cost-effectiveness over a much shorter, 10 year, time horizon (Supplementary Table S31), giving an ICER of $618, although here, as a result of the truncated time horizon,

the sensitivity analysis in which we calculated DALYs including years of life lost beyond the end of the time horizon gave an ICER of $216.

## Characteristics of setting-scenarios predicting cost-effectiveness

We fitted a series of logistic regression models across setting-scenarios to evaluate characteristics of setting-scenarios in 2024 predicting cost-effectiveness of the intervention (Supplementary Table S32). The strongest predictor is the percentage of diagnosed PWH with VL < 1000 copies/mL. As shown in Table 5, the ICER increases from $79 when the percentage of diagnosed PWH with VL > 1000 copies/mL is ≥20% to $627 when the percentage is below 7%. Table 5 also shows sensitivity analyses according to prevalence, incidence, the percentage of PWH diagnosed, the percentage of diagnosed PWH on ART, and the percentage of PWH on ART who have viral load <1000 cps/mL.

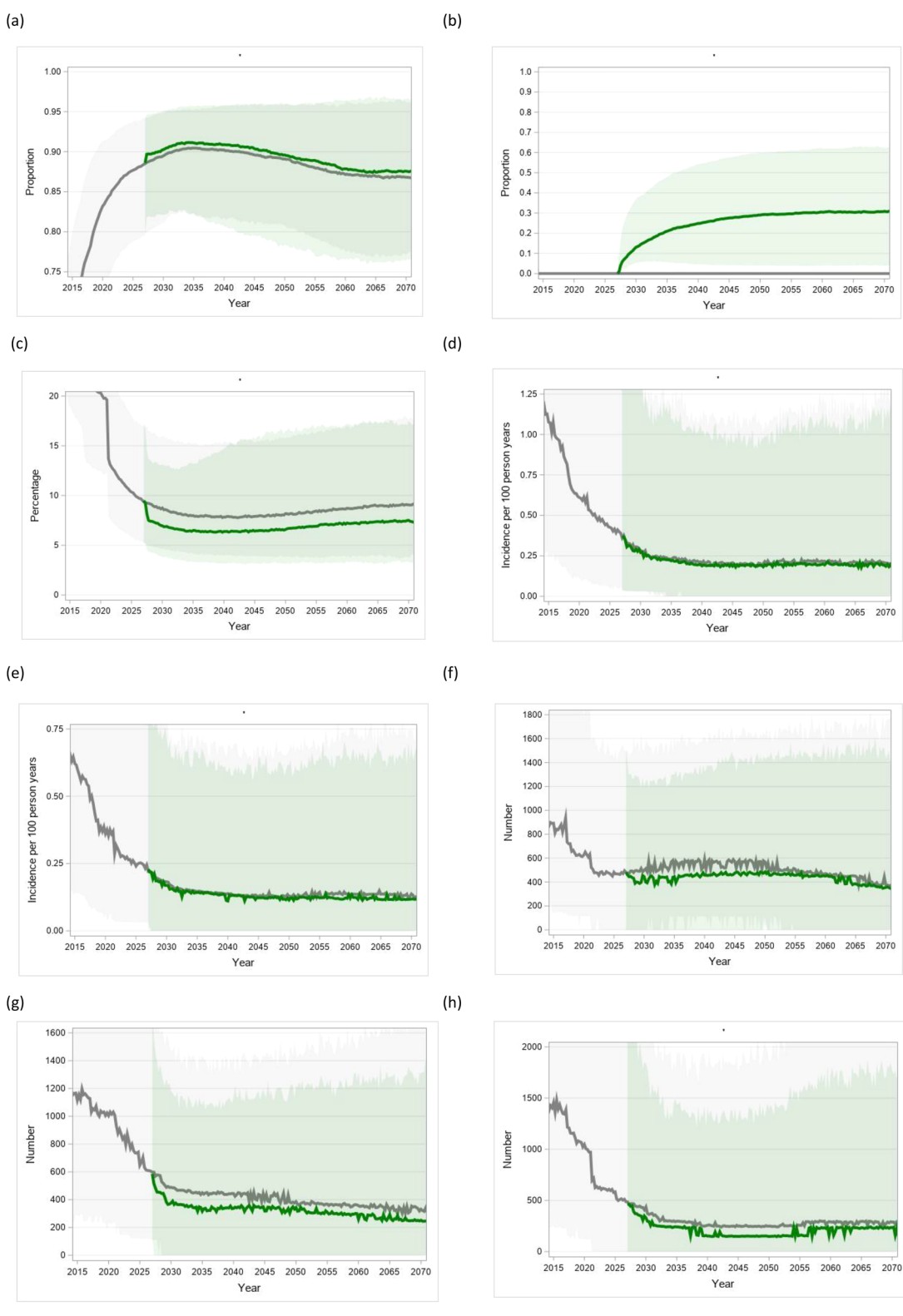

No lenacapavir + cabotegravir introduction ——— Lenacapavir + cabotegravir introduction ———

**Fig. 1 | Effects over time of introduction of lenacapavir + cabotegravir. a** Of PWH, percentage on ART (**b**) Of PWH on ART, percentage on lenacapavir + cabotegravir (**c**) Of PWH with diagnosed HIV, percentage with viral load >1000 cps/mL (**d**) HIV incidence in women aged 15−49 (**e**) HIV incidence in men aged 15−49 (**f**) Number of HIV-related deaths in women ^ (**g**) Number of HIV-related deaths in men (in the context of a setting with population size of 1 million adults aged 15+ in 2024) (**h**) Number of children newly infected with HIV per year (in the context of a setting with population size of 1 million adults aged 15+ in 2024).

**Table 4 | Breakdown of costs. Discounted annual costs in $m over 50 years ^**

| | No lenacapavir + cabotegravir introduction | Lenacapavir-cabotegravir introduction |
|---|---|---|
| ART drug (len-cab) | 3.35(0) | 3.71 (1.29) |
| Cotrimoxazole | 0.27 | 0.27 |
| ART clinic visits | 1.23 | 1.73 |
| Viral load tests | 1.13 | 1.15 |
| CD4 count tests | 0.05 | 0.05 |
| Clinical disease care costs HIV-related | 0.62 | 0.53 |
| Pre-death care (non-HIV) | 0.26 | 0.26 |
| HIV testing | 1.52 | 1.51 |
| PrEP drug | 0.67 | 0.65 |
| PrEP clinic visits | 0.58 | 0.57 |
| VMMC | 0.36 | 0.36 |
| Condom availability | 0.71 | 0.72 |
| Care for children with HIV | 0.31 | 0.23 |
| Total | 11.07 | 11.74 |

^ In the context of a setting with population size of 1 million adults aged 15+ in 2024; for a setting with x.x million adults the number can be multiplied by x.x.

### Sensitivity analyses around implementation of policy and costs

In our main analysis reported above the total cost of lenacapavir + cabotegravir drug plus clinic visits is $140 per year ($80 drug cost plus $60 clinic costs). If instead these annual costs were $180 then the ICER becomes $547 per DALY averted, and $814 if the annual cost were $220. On the other hand, at an annual cost of $100 the ICER is $12 per DALY averted. Further, the ICER increases from $206 to $434 as the percentage of PWH on lenacapavir + cabotegravir who started when already virally suppressed or when drug naïve (mean over 10 years) increases from below 20% to above 60%. The ICER also decreases as the percentage of PWH who started lenacapavir + cabotegravir when on ART with viral non-suppression increases (Table 5), but does not depend on the percentage of PWH on lenacapavir + cabotegravir who started when off ART or the overall scale of lenacapavir + cabotegravir uptake.

### Alternative policies for introduction of lenacapavir + cabotegravir

We also consider possible alternative policies for introduction of lenacapavir + cabotegravir (Table 6).

If lenacapavir + cabotegravir is restricted to women aged 15–39 then the ICER across settings is $138 per DALY averted. If lenacapavir + cabotegravir is restricted to adolescents and young PWH aged 15–24, due to the higher levels of poor adherence at these ages, then the ICER across settings is $18.

In Fig. 2 we summarize the ICERs for introduction of lenacapavir + cabotegravir under various baseline conditions.

## Discussion

Our modelling analysis presented here suggests that there is potential for substantial health benefits of introduction of lenacapavir + cabotegravir long-acting treatment were it to be introduced in ECSWA settings. Across all setting scenarios, our assumptions on uptake led to a median 15% of PWH on ART being on lenacapavir + cabotegravir over the first 10 years from its introduction. Given this, there was a decrease in median percentage of diagnosed PWH with viral load >1000 copies/mL by 1.4% from 8.3% to 6.9% over the first 10 years,

and this was predicted to lead to substantive decreases in HIV deaths, HIV incidence and mother to child transmission of HIV. We found that cost-effectiveness of lenacapavir + cabotegravir introduction for a given setting, as well as its 10 year impact on mortality, is largely determined by the percentage of diagnosed PWH virally suppressed on ART (Supplementary Tables S30 and S32). If lenacapavir + cabotegravir can be delivered at a total cost per person of $140 per year ($80 for drug and $60 for delivery of injections) then it is likely to be cost-effective if the percentage of diagnosed PWH virally suppressed on ART is below 93%. Based on the most recent PHIA and other survey data, this would be the case, for example, in Zimbabwe, Tanzania, Ethiopia, Mozambique, Uganda, Lesotho and South Africa. If lenacapavir + cabotegravir can be delivered for $100 per year then it is likely to be cost-effective in almost all settings. We find an immediate impact on HIV deaths with a relatively small short term budget impact of a 3.5% rise. We suggest that implementation studies be conducted to explore the viability of these approaches.

Long-acting treatment has potential benefits for people unable to adhere to daily oral treatment. Reasons for such challenges might include difficulty with keeping or carrying pills for various reasons, including external and internal stigma, food insecurity, beliefs in alternative medicines, mental health challenges, and alcohol misuse[29–32]. There has been high interest expressed by communities affected by and living with HIV for access to long-acting injectable treatment for HIV[4–9]. This is supported by experience with long acting injectable contraception, which is the most used modern method of contraception in much of Africa[33]. We have proposed that use of lenacapavir + cabotegravir is prioritized towards those on oral drugs who are unable to maintain sufficient adherence and those who are not under care for their HIV due to such challenges. The targeting of long-acting ART to those with adherence challenges has precedence with long-acting cabotegravir + rilpivirine[11–15]. We hypothesize that active offer to people living with diagnosed HIV who are not in care will bring some people back into care. Those that have stopped ART due to adherence challenges might also present with advanced HIV disease and long acting treatment might be of particular benefit for this group. However, we also recognise that this raises questions of equity and suggest that those not fitting with these criteria, including those with viral suppression on oral drugs, not be excluded from access if they express a strong preference for long-acting treatment as this could create an incentive for non-adherence. It could also be that in such people there is a high risk of future non-suppression. We showed that the cost effectiveness of lenacapavir + cabotegravir introduction is reduced with an increasing proportion of people who were switched to the regimen despite having viral load suppression on oral therapy. One aim for implementation studies would be to ascertain which approaches are realistic and scalable. We also considered alternative introduction criteria, such as to women age 15–40 only, but these would be difficult to implement and we favour as broad access as possible where there are likely to be benefits.

Enough studies have shown success in offering long-acting injectable ART to those with viral non-suppression (including in homeless populations in the United States)[10–13], that cabotegravir + rilpivirine has been endorsed by major U.S.-based guidelines in those without virologic suppression and adherence challenges[22,23]. As mentioned above, a case series of people with HIV on the lenacapavir + cabotegravir has shown promise for this regimen[28]. Cabotegravir is viewed as safe at the time of conception and during pregnancy[34]. While data for lenacapavir are limited (e.g. 193 pregnancies in a recent study of lenacapavir as prevention[25]) there have not been safety issues identified to date. There are however drug interactions with rifampicin for both cabotegravir and lenacapavir resulting in lowered levels of the antiretrovirals (https://www.hiv-druginteractions.org/)[35].

A minority of people who have viral non-suppression on a dolutegravir-based regimen will carry virus with resistance to

**Table 5 | HIV-related deaths, DALYs and costs over 50 years; cost-effectiveness analysis. Values are means over setting-scenarios**

| | No lenacapavir + cabotegravir introduction | Lenacapavir + cabotegravir introduction |
|---|---|---|
| Difference in number of HIV-related deaths per year ^ | --- | -195 |
| Difference in DALYs per year ^ | --- | -2400 |
| Difference in annual cost ^ | --- | +$0.67 m |
| Difference in net DALYs per year ^ | --- | -1060 |
| Incremental cost-effectiveness ratio | --- | $ 280 |
| Percent of setting scenarios for which policy incurs the lowest DALYs | 18% | 82% |
| Percent of setting scenarios for which policy has the lowest cost | 85% | 16% |
| Percent of setting scenarios for which policy has the lowest net DALYs (i.e. it is cost-effective) | 39% | 61% |
| Incremental cost-effectiveness ratio according to total annual cost of lenaca-pavir + cabotegravir and clinic costs for delivery | | |
| $ 100 | --- | $ 12 |
| $ 140* | --- | $ 280 |
| $ 180 | --- | $ 547 |
| $ 220 | --- | $ 814 |
| * cost used in primary analysis | | |
| Incremental cost-effectiveness ratio according to the percentage of PWH on lenacapavir + cabotegravir who started when already virally suppressed (mean over 10 years). | | |
| <20% | --- | $ 206 |
| 20%–39% | --- | $ 233 |
| 40%–59% | --- | $ 328 |
| ≥60% | --- | $ 434 |
| Incremental cost-effectiveness ratio according to the percentage of PWH on lenacapavir + cabotegravir who started when on ART with viral non-suppression (mean over 10 years). | | |
| <15% | --- | $ 466 |
| 15%–24.9% | --- | $ 293 |
| 25%–44.9% | --- | $ 197 |
| ≥45% | --- | $ 143 |
| Incremental cost-effectiveness ratio according to the percentage of PWH on lenacapavir + cabotegravir who started when off ART (mean over 10 years). | | |
| <30% | --- | $ 311 |
| 30%–44.9% | --- | $ 265 |
| 45%–59.9% | --- | $ 254 |
| ≥60% | --- | $ 280 |
| Incremental cost-effectiveness ratio according to the percentage of PWH on ART who are on lenacapavir + cabotegravir (mean over 10 years). | | |
| <10% | --- | $ 128 |
| 10%–14.9% | --- | $ 281 |
| 15%–19.9% | --- | $ 247 |
| 20%–24.9% | --- | $ 241 |
| >25% | --- | $ 389 |
| Incremental cost-effectiveness ratio according to HIV prevalence age15–49 | | |
| <5% | --- | $ 448 |
| 5%–9.9% | --- | $ 316 |
| 10%–14.9% | --- | $ 339 |
| 15%–19.9% | --- | $ 209 |
| 20%–24.9% | --- | $ 236 |
| ≥ 25% | --- | $ 270 |

**Table 5 (continued) | HIV-related deaths, DALYs and costs over 50 years; cost-effectiveness analysis. Values are means over setting-scenarios**

| | No lenacapavir + cabotegravir introduction | Lenacapavir + cabotegravir introduction |
|---|---|---|
| Incremental cost-effectiveness ratio according to HIV incidence (/100 person years) age 15–49 | | |
| <0.15 | --- | $ 390 |
| 0.15 – 0.29 | --- | $ 450 |
| 0.30 – 0.44 | --- | $ 316 |
| 0.45 – 0.60 | --- | $ 331 |
| 0.60 – 0.75 | --- | $ 305 |
| ≥ 0.75 | --- | $ 213 |
| Incremental cost-effectiveness ratio according to the percentage of PWH diagnosed | | |
| < 80% | --- | $ 214 |
| 80%–84.9% | --- | $ 288 |
| 85%–89.9% | --- | $ 265 |
| ≥ 90% | --- | $ 288 |
| Incremental cost-effectiveness ratio according to the percentage of diagnosed PWH on ART | | |
| < 85% | --- | Cost-saving |
| 85%–89.9% | --- | $ 197 |
| 90%–94.9% | --- | $ 253 |
| ≥ 95% | --- | $ 327 |
| Incremental cost-effectiveness ratio according to the percentage of PWH on ART with VL < 1000 cps/mL | | |
| <85% | --- | Cost-saving |
| 85%–89.9% | --- | $ 96 |
| 90%–94.9% | --- | $ 221 |
| ≥ 95% | --- | $ 456 |
| Incremental cost-effectiveness ratio according to the percentage of diagnosed PWH with VL > 1000 cps/mL | | |
| <7% | --- | $ 627 |
| 7%–9.9% | --- | $ 457 |
| 10%–14.9% | --- | $ 288 |
| 15%–19.9% | --- | $ 157 |
| ≥ 20% | --- | $ 79 |

^ In the context of a setting with population size of 1 million adults aged 15+ in 2024; for a setting with x.x million adults the number can be multiplied by x.x. All outputs refer to adults age 15+ unless otherwise stated.

dolutegravir. In such cases, a move to a lenacapavir + cabotegravir regimen would not be advisable due to cross resistance between cabotegravir and dolutegravir meaning that co-occurring resistance to cabotegravir is likely[36]. Using a two -drug regimen that has resistance to cabotegravir could expose lenacapavir as monotherapy which would likely eventually lead to resistance to lenacapavir[37,38]. Ideally, a resistance test would be performed before such a switch is considered but such testing is not generally available for individual person management in ECSWA. Our modelling takes this effect into account and despite the small risk of this occurring we found overall strong net benefits. In practice, the switch to lenacapavir + cabotegravir should only be made in people for whom ongoing adherence is poor, as resistance is very unlikely to be present without selective drug pressure. While self-report of adherence is known to be unreliable, self-report of non-adherence is likely to reflect true non-adherence. If not already done as standard in a person starting a new regimen, an additional consideration might be to make a viral load measure 4–6 months after the start of lenacapavir + cabotegravir to check that viral suppression has occurred, in addition to standard annual

**Table 6 | Effects of alternative policies for introduction of lenacapavir + cabotegravir**

| | No lenacapavir + cabotegravir introduction | Lenacapavir + cabotegravir introduction unrestricted by age or sex*^ | Lenacapavir + cabotegravir introduction for women aged 15–39^ | Lenacapavir + cabotegravir introduction for adolescents and young people aged 15–24^ |
|---|---|---|---|---|
| Difference in number of HIV-related deaths per year ^^ | ... | -195 | -70 | -45 |
| Difference in DALYs per year over 50 years compared with no lenacapavir + cabotegravir ^^ | ... | -2400 | -1170 | -660 |
| Difference in annual cost over 50 years compared with no lenacapavir + cabotegravir ^^ | ... | +$ 0.67 m | +$ 0.16 m | -$0.01 m |
| Difference in net DALYs per year over 50 years compared with no lenacapavir + cabotegravir ^^ | ... | -1060 | -840 | -630 |
| Cost per DALY averted compared with no lenacapavir + cabotegravir introduction | ... | $ 280 | $ 138 | $ 18 |
| Percent of setting scenarios for which policy incurs the lower net DALYs than no lenacapavir + cabotegravir. | ... | 61% | 60% | 59% |

* this is the primary analysis as in Table 5. ^ In all three policies there remains the prioritization of offer to people with unsuppressed HIV and people off ART. ^ In the context of a setting with population size of 1 million adults aged 15+ in 2024; for a setting with x.x million adults the number can be multiplied by x.x. All outputs refer to adults age 15+ unless otherwise stated.

monitoring. Any such additional cost would be envisaged as being within the $60 per year clinic costs.

The fact that currently cabotegravir requires a 2 monthly intramuscular injection in the buttock and lenacapavir a 6 monthly subcutaneous injection in the abdomen means that the regimen of the two drugs is not synchronously administered. This, and the dependence of people attending when an injection or both injections are due raises particular challenges in ensuring that injections for both drugs are delivered on time. If not, this could lead to low levels of one or both drugs. Data from studies of drug concentrations beyond 2 months have led to recommendations that with a 2 month delay in cabotegravir injection then drug concentrations remain sufficient such that injections can resume without the need for a new loading dose[39,40]. Newer formulations of cabotegravir might allow for dosing every 3 or 4 months[41], making it more convenient for synchronized administration with lenacapavir and reducing the requirement for clinic visits from six per year to three or four per year. Lenacapavir also currently requires an oral lead-in dose.

Implementation studies are needed to further understand whether introduction of lenacapavir + cabotegravir and, in particular, the approach we have suggested has potential. It is important that there is community engagement at every stage, especially since this might help to manage and address some of the equity issues but also prepare, raise awareness and incentivize targeted PWH eligible to access long acting injectable treatment. Implementation studies might initially recruit people who are attending clinic but self-report poor adherence to oral medication and have unsuppressed viral load. As more experience is gained and if and when it becomes clear that the injections can be consistently delivered in clinics and viral suppression attained in such PWH then this would provide evidence to support roll out of offer of lenacapavir + cabotegravir in this group. Implementation studies might then move on to studies which seek out people who have dropped out of care to offer them the option of lenacapavir + cabotegravir. This would initially be in people who are unlikely to be mobile who could likely be found if they did not return to care after their first injection in order to ascertain reasons. As more experience is gained it may be possible to explore community-based delivery of injections. Frequent viral load monitoring, ideally at point of care, and resistance testing will be important in such studies to check that the regimen is leading to sustained viral suppression and not associated with development of drug resistance mutations. Implementation studies can learn from similar such studies conducted in high income settings among those with adherence challenges[10]. Implementation studies will also inform the long term tolerability of the lenacapavir + cabotegravir regimen outside of clinical trial settings, and in particular over effects of injection pain and nodules. Additional implementation studies could explore if offering an injection to people who are just initiating ART, particularly those with advanced HIV disease, would lead to improved retention in care in the critical first 6 months of treatment. There is also the concern over hepatitis B as, unlike tenofovir, lenacapavir and cabotegravir are not active against hepatitis B. Implementation studies could inform future use of long acting injectable treatment even if lenacapavir + cabotegravir is ultimately not the regimen that is scaled up. At this stage, we recommend that policy-makers track the implementation studies of the introduction of long-acting injectable treatment and begin to consider practical approaches to its introduction.

Other potentially cost-effective innovations proposed to enable people to stay on treatment with viral suppression have been proposed, such as making the oral daily ART regimen of TLD available in communities at no cost, for use as treatment for those living with HIV who are without their drug as well as for post-exposure prophylaxis to prevent HIV, which requires urgent initiation after risky condomless sex[42]. These two approaches address different adherence challenges and may have complementary benefits.

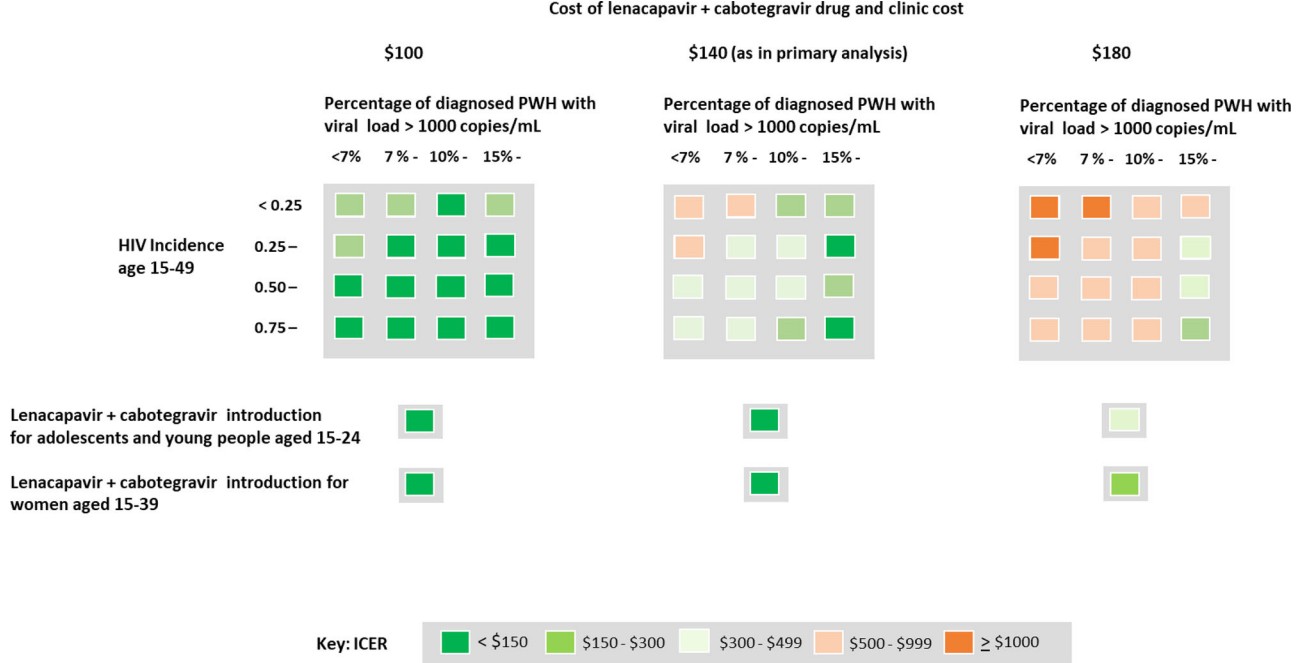

**Fig. 2 | Incremental cost-effectiveness ratio for lenacapavir + cabotegravir introduction according to the percentage of diagnosed PWH with viral load >1000 copies/mL and HIV incidence age 15–49.** Also shown is the incremental cost-effectiveness ratio for lenacapavir + cabotegravir introduction in subgroups defined by age and sex. Both are also shown according to cost of lenacapavir + cabotegravir.

We considered DALY benefits of introduction of long acting lenacapavir + cabotegravir but we did not include positive effects on quality of life due to the potentially reduced stigma and lack of stress of having to have oral pills available every day. Thus we may have under-estimated the full benefits. There is, however, also potentially stigma created with injectables via the need for more frequent clinic visits and from the subcutaneous nodules that can persist for many months.

There is potential for both cabotegravir and lenacapavir to increasingly be used as pre-exposure prophylaxis (PrEP) as individual agents to prevent acquisition of HIV[25,43] although future accessibility of these is uncertain. Both have been shown to have extremely high efficacy[25,43]. Use of cabotegravir as PrEP, however, has the potential to increase levels of integrase inhibitor drug resistance[44]. In our modelling we considered the possibility of either low or high future uptake of cabotegravir PrEP and the effects of use of cabotegravir in a treatment regimen and on resistance to this drug on its future efficacy as PrEP. Future lenacapavir use as PrEP was not included. In addition to considering the effect on treatment effectiveness of use of cabotegravir and lenacapavir as PrEP, it is relevant also to consider the inverse, whether use of lenacapavir + cabotegravir as treatment could undermine their effectiveness as PrEP due to development of drug resistance and the fact that PrEP would not be effective against virus with drug resistance to the PrEP drug. Explicit future modelling of this question is planned.

Costs with which lenacapavir + cabotegravir can be sourced and delivered to people with HIV is currently very uncertain. It has been suggested that there is potential for lenacapavir to be produced at around $40 per year with wide scale demand and cabotegravir below $20 per year[45,46]. Viiv have awarded voluntary licences to three companies for production of cabotegravir, and Gilead Sciences recently announced that six generic manufacturers have been awarded voluntary licenses for the production of lenacapavir[47] for both treatment and prevention in most African countries. Cost will be lower with greater volume and up-front commitment to buy large volumes. If the

drugs were to be used by many people with existing viral suppression it could lead to lower prices.

Although long-acting treatment offers a potential option for people who find difficulties with maintaining daily pill taking, it does require close follow-up to ensure that injections are delivered on time. This would put pressure on clinics and would require adaptations. Follow up may be incorporated in existing differentiated service delivery models of care or via community workers. We conservatively assumed an annual cost of $60 per year per recipient of care for injection delivery, compared with $20 per year for a person with viral suppression on oral ART, to account for the greater intensity of activity in clinics but implementation studies will be needed to better inform this. Administering injections will be more time consuming and more difficult to deliver than simply dispensing drugs although it may be in future that the injections can be undertaken by community health workers or even with self-injection or injection done by a family member or friend, as occurs for long acting contraceptives (https://www.afro.who.int/countries/burkina-faso/news/self-injectable-contraception-successes). Current adherence counselling messages will be replaced with messages regarding the importance of being on time for injections and efforts to trace people who do not adhere to their appointments will still be required. However, there will be no more guessing about a person's true adherence in the presence of an unsuppressed viral load.

In some settings if donor support is not available or reduced from current levels it may be that a cost-effectiveness threshold of around $150 is relevant rather than the $500 we used[48], in which case a lower cost of lenacapavir + cabotegravir will be required. If targeted at women aged 15–39 or young people aged 15–24 the introduction is predicted to be cost-effective with a drug cost of $80 even in the context of a cost-effectiveness threshold of $150.

Clinic costs for lenacapavir + cabotegravir remain especially uncertain and might fall with innovations in delivery. Many countries currently have severely constrained health spending and consideration will need to be given to financing and affordability. If

lenacapavir + cabotegravir can be introduced without substantially adding to clinic visit costs, it is very likely to be cost-effective. The budget impact of introduction of lenacapavir + cabotegravir at the costs assumed would be an increase of $0.67 million per 10 million adults per year, an increase of 3.5%. This is a relatively modest amount when weighed against the costs of optimized client centered programming.

A limitation of our work is that while data have suggested great potential for long-acting injectable treatment in Africa, there are as yet no pilot data showing uptake of lenacapavir + cabotegravir or implementation study data on injection delivery of this regimen, which at this point requires 2 monthly injections of cabotegravir and 6 monthly injections of lenacapavir. There are as yet only very limited clinical data on use of the regimen in any part of the world[28] and it has not been approved as a combination regimen so further pilot data are needed. Further, unlike the current first line oral HIV treatment regimen, this injectable regimen does not provide treatment for active hepatitis B infection. Given uncertainty, we assume a wide range of levels of interest in long-acting injectable treatment, and it is possible that this is overestimated. Persistence with on-time cabotegravir and lenacapavir injections, as well as ease of transferring to other clinics without treatment interruption during times of mobility is also uncertain. With the above mentioned exceptions there is generally a wide range of data to inform all aspects of our model but we recognise that more extensive data would always be useful. Lastly, while there are benefits of modelling a range of setting scenarios representing the diversity of settings in ECSWA in that we can assess what attributes of a setting influence impact and cost-effectiveness, we recognise that we need to take care in interpretation of outcomes for an "average" setting scenario and that there would be some additional value in modelling the effect of lenacapavir + cabotegravir implementation in the context of a model calibrated to data from one or more specific countries in aggregate.

In conclusion, we find that in epidemic settings where viral suppression levels in diagnosed PWH is sub-optimal there is potential for introduction of long acting injectable treatment to have a significant beneficial impact on HIV mortality in ECSWA and to be cost-effective. Pilot implementation studies are needed to further understand whether the approach has potential.

## Methods

### HIV synthesis model
We have previously described our HIV Synthesis model and, for example, how it was applied to considering risks and benefits of dolutegravir introduction in combined antiretroviral regimens[49–51], the introduction of cabotegravir as pre-exposure prophylaxis (PrEP)[32] and cabotegravir-rilpivirine as treatment[24]. Full details are given in the Supplementary Model Details. Each run of the simulation programme creates 100,000 simulated people who will be age 15 or above at some point between 1989 (taken as the start of the HIV epidemic) and 2076 (see sections 1 and 2 of the Supplementary Model Details). Table S27 in the Supplementary Model Details describes parameters and the distributions that parameter values are sampled from. Variables defined for each individual and updated every 3 months, include age, sex, primary and non-primary condomless sex partners, whether currently a female sex worker, HIV testing, male circumcision status, presence of sexually transmitted infections other than HIV, and use of oral and, from 2027 the possible scale up of cabotegravir as PrEP. Only heterosexual sex is modelled. Three possible future trajectories of population growth are considered (see sections 1 and 2 of the Supplementary Model Details). The initial age distribution for both males and females is sampled for each population simulation from three possible distributions representing these different population demographic structures. These are chosen such that in the absence of HIV, and given the death rates, the resulting population pyramids and

growth rates represent the range of those seen across the setting scenarios. Thus a proportion of simulated people have an age below 15 in 1989 (and most are yet to be born). The only variable that is modelled and updated up to reaching the age of 15 (when becoming potentially sexually active) is age itself. This results in a simulated adult population size in 2024 of a median of 31,626 (90% range 25,225 to 35,522)

In HIV-positive people, we model viral load, CD4 cell count, use of specific antiretroviral drugs and presence of specific drug resistance mutations. Risk of AIDS death in the model depends on the current CD4 cell count, viral load, age and ART status. For a person on treatment the viral load, CD4 cell count and risk of resistance are primarily determined by the adherence / drug concentration and the number of active drugs being taken. The activity level of each drug depends on its underlying potency and which, if any, drug resistance mutations are present. Informed by short term viral suppressive capacity as monotherapy, we assume that nucleoside reverse transcriptase inhibitors (NRTIs) 3TC and tenofovir have potency 1 and dolutegravir and darunavir have a potency of 2.

Through our sampling of parameter values (see Supplementary Model Details) at the start of each model run we create 1000 "setting-scenarios" reflecting uncertainty in assumptions and a range of characteristics similar to those seen in ECSWA. For each run we sample parameters from the same distributions. These represent sub-settings within countries as well as countries as a whole. This approach means we can evaluate baseline characteristics of a setting which predict cost-effectiveness of an intervention, allowing us to understand how results generalize across the region. For each setting scenario when we present absolute numbers of health-related events, costs and DALYs we scale-up our simulated population by multiplying by a setting-scenario-specific scale factor so that our results are expressed per 1 million adults age 15+ (in 2024).

### Introduction of lenacapavir + cabotegravir treatment
We assume that the policy of introduction of lenacapavir + cabotegravir treatment would involve active offer of lenacapavir + cabotegravir in people with sustained viral load measured >1000 copies/mL (despite enhanced adherence counselling) on oral drugs, and to people living with diagnosed HIV who are not currently engaged in treatment; i.e. contacting or visiting people who have previously been in care or are known to be diagnosed but never started ART to see whether the offer of lenacapavir + cabotegravir encourages them to re-start ART. The policy would also involve some switching from oral drugs to lenacapavir + cabotegravir in people with ongoing viral suppression with oral drugs who express a strong preference for lenacapavir + cabotegravir. The extent of the uptake of lenacapavir + cabotegravir by each of these groups is determined by sampling relevant parameters for each setting-scenario as described below and in the Supplementary Model Details. It is felt that it cannot be an absolute condition for lenacapavir + cabotegravir access that the viral load is unsuppressed as that could become an incentive to interrupt oral drugs.

A person on long acting injectable cabotegravir or lenacapavir is assigned as having 100% of the drug concentration required, equivalent to 100% daily pill taking adherence to an oral drug regimen, for the recommended period of time between doses. For lenacapavir this is every 6 months. The current approved dosing for cabotegravir is every 2 months. Since our model operates with a 3 month time step we make the simplifying assumption that the drug concentration of cabotegravir for a person on cabotegravir is 100% if an injection was received in a given 3 month period.

We assume cabotegravir has a potency of 2 (Supplementary Model Details) while lenacapavir has a potency of 1.5 (25%) / 2.0 (75%) (choice of value determined for each model run by sampling), based on its effect in highly treatment experienced people[26,27]. For people on

ART there is the chance of interruption, which is determined by an underlying parameter value selected at the start of each model run, the relative rate of interruption for those on lenacapavir-cabotegravir compared with those on oral drugs, and modified by individual person characteristics, such as whether they have a current toxicity to their drugs (including injection site pain/nodules for long acting drug). If lenacapavir + cabotegravir is stopped and no oral drugs started then there is an increased risk of lenacapavir resistance in months 3–6 after the stop due to lenacapavir being effectively a monotherapy for this period.

Parameters relating to lenacapavir + cabotegravir introduction include *lencab_uptake_vlg1000* is the probability that a person who has a measured viral load above 1000 copies/mL despite enhanced adherence advice is offered and accepts to start lenacapavir-cabotegravir. This applies to each time a person on oral drugs has a new viral load value >1000 copies/mL so long as it is at least 1 year since the last offer. *lencab_uptake* is the probability per 3 months that a person who has not been identified as having an indication for lenacapavir-cabotegravir nevertheless starts lenacapavir-cabotegravir due to having a strong preference. *prob_strong_pref_lencab* is the proportion of people who will have a strong preference for lenacapavir-cabotegravir even if they are able to be highly adherent to oral drugs. *rate_return_for_lencab* is the probability that a person with diagnosed HIV who is out of care returns to care and starts lenacapavir-cabotegravir as a result of clinic outreach with the offer of lenacapavir-cabotegravir. As for anyone on ART there is viral load monitoring in place (with probability of a viral load test being performed when indicated determined by parameter *prob_vl_meas_done*) and people can have two consecutive values above 1000 copies/mL (if resistance has emerged) which leads to switching back to an oral regimen. People without viral non-suppression can also switch to oral drugs; a parameter *rate_lencab_to_tld* determines the probability that a person on lenacapavir-cabotegravir switches back to oral drugs. Given the lack of experience with introduction of long acting injectable drugs for HIV with criteria for targeting such as we propose in ECSWA settings we sample these parameters from wide distributions to reflect uncertainty and to allow as to study the relationship between uptake in certain sub-populations with cost-effectiveness.

### Cost-effectiveness analysis

DALYs are calculated in the standard way as the sum of years of life lost due to premature death and years lived with disability (using the weights shown in Supplementary Model Details Table S28). Years of life lost as a result of each death are counted only until the end of the time horizon for analysis. In a sensitivity analysis we also present results where years of life lost as a result of each death are extended fully over the expected years of life remaining at death, even if these go beyond the end of the time horizon, as is commonly done.

Cost- effectiveness analysis is conducted from a healthcare perspective. Costs and health outcomes were both discounted to present US$ values at 3% per annum, and a cost-effectiveness threshold of US $500 per DALY averted was used. Country-specific thresholds are uncertain but $500 averted per DALY averted is likely to be at the upper end on the basis of evidence concerning how resources would otherwise be used[48,52], particularly in the new funding environment in which PEPFAR support is reduced. We used this threshold to calculate net DALYs averted[53]. Net DALYs take into account the health consequences of the difference in costs, for a given cost-effectiveness threshold, as well as the difference in health (DALYs) and reflect the impact of a policy on overall population burden of disease: Net DALYs averted = DALYs averted + difference in costs/cost effectiveness threshold. We model for each woman pregnancies, births and periods of breastfeeding, with the probability of transmission dependent on the mother's viral load (see Supplementary Model Details). For each child infected through mother to child transmission we assume that 5

DALYs (with discounting) are incurred. This is likely an under-estimate of the DALYs incurred but we wished to err on the side of conservatism in our evaluation of the benefits of long-acting treatment. While it is accepted that cost-effectiveness analysis should adopt a suitably long time horizon to fully account for all benefits and risks/harms with alternative policies, we present results from a sensitivity analysis in which we use a 10 year time horizon instead of 50 years.

Costs for tenofovir-lamivudine-dolutegravir are assumed to be $ 50 per year including supply chain costs[54]. Clinic costs for people on oral drugs are assumed to be $ 10 per 3 months if the person is not known to have viral suppression and $ 5 per 3 months if the person has had a recent viral load measure showing viral suppression[55]. Drug costs at scale for implementation for lenacapavir + cabotegravir are unknown at this point although estimates have been made[45,46]; we initially use a placeholder cost of $ 80 per year including supply chain costs and then show the effects of variation in this. Similarly with clinic costs for people on lenacapavir + cabotegravir, we initially use a cost of $ 15 per 3 months (so three times higher than for a person on oral drugs with current documented viral suppression). The lack of need for daily drug adherence counselling and the possibility of administration of injections in communities could mean that such costs become lower, although there is co-administration of two separate products. The lifetime healthcare cost incurred, with discounting, as a result of each child born with HIV is assumed to be $ 1000. Again, this is likely an under-estimate but we wished to err on the side of conservatism in our evaluation of the benefits of long-acting treatment. Other costs and disability weights are shown in the Supplementary Model Details.

The model is coded in SAS 9.4.

This modelling study did not require ethical approval.

### Reporting summary

Further information on research design is available in the Nature Portfolio Reporting Summary linked to this article.

## Data availability

This modelling study is based on simulations and there is no analysis of empirical data. Model parameters are included in the Supplementary Information.

## Code availability

The code is available on figshare: https://doi.org/10.6084/m9.figshare.28083098.v1 and https://doi.org/10.6084/m9.figshare.28401485.v1.

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

## Acknowledgements
This study was made possible by an award from the Gates Foundation to Andrew Phillips. INV-007145.

## Author contributions
Consideration of the concept and input into modelling approach, critical comments on results and manuscript draft, suggested edits, final approval of manuscript: all authors. Coding of the model: A.P., L.B.-M., and J.S. Initial draft of manuscript and finalisation of manuscript: A.P.

## Competing interests
D.H.: for the NIH funded SEARCH study, ViiV provided cabotegravir for the study. C.F. served as an expert witness for a law firm representing Gilead in a patent dispute in 2023. Payments came from the law firm and not from Gilead, and the payments were not contingent on outcome. The remaining authors declare no competing interests.
