## [Transparent Peer Review file · Nature Communications]

Potential impact and cost-effectiveness of long-acting injectable lenacapavir plus cabotegravir as HIV treatment in Africa

Corresponding Author: Professor Andrew Phillips

Version 0:

Reviewer comments:

Reviewer #4

(Remarks to the Author)

Manuscript: Enabling higher levels of HIV viral suppression in Africa: A modelling study of the potential impact and cost-effectiveness of introduction of long-acting injectable lenacapavir plus cabotegravir (NCOMMS-25-35509-T)

Review date: 5/2025

SUMMARY

This manuscript describes a modeling analysis of the potential health and economic implications of using long-acting ARV agents (in place of current regimens) as part of routine ART in Sub-Saharan African countries. I have focused my review on a set of outstanding issues that remain from earlier reviews. These four points are referred to below, in the order presented in the most recent response to reviewers comments.

ISSUE #1: Relevance of cost-effectiveness results estimated over a 10-year analytic horizon.

It is conventional for cost-effectiveness analyses to estimate results over a future period that is longer enough such that all (or at least the large majority) of health and economic consequences of the intervention are represented in the analysis. For most health interventions a lifetime horizon is the default option. Where an intervention impacts disease transmission this complicates matters, but the 50-year horizon adopted by this analysis appears sufficient to represent the large majority of consequences (and discounting means that differences in outcomes beyond 50 years will have a minimal impact). I do not agree with the position that 10-year outcomes will be more salient for policy-makers (with the implication that later outcomes should be ignored). There are a large number of health interventions that are widely accepted to be high value and that require up-front spending to produce health gains that are realized over the remainder of the individual's lifetime. I worry that additionally presenting the 10-year CEA results could be misleading for readers who may not appreciate the distinctions. That said, one thing that would be useful to communicate to readers is the relative timing of costs and health benefits. This is partly achieved by Figure 1. I would suggest the authors review the results/discussion text to check whether there are edits that could be made to communicate these dynamics clearly.

ISSUE #2: Description of the study population.

I agree with the reviewer that the age of modelled individuals needs to be stated clearly in the main text. In the methods section I see that the authors refer to individuals "age 15 or above". If this is correct then I think that is sufficient. The text that has been added which refers to individuals of "age -72" (line 647) worries me a bit, I feel this will confuse readers rather than clarify, and suggest omitting. Elsewhere in the manuscript reference is made to individuals aged 15-49. I understand this to be due to the need to compare model estimates to external estimates that adopt this age range, and think this is reasonable.

ISSUE #3: Which (if any) countries does the analysis represent.

The point raised by the reviewer questions whether the model is a sufficient representation of individual countries. This is an important point, as the policy question addressed by the study has to be relevant to individual countries to be useful. The

authors took the approach of defining a small number of model inputs that represent differences between settings, creating ranges for these variables (representing extreme low and high values for what might be expected in any country or sub-national setting), and then randomly sampling from these ranges to create a bank of results in which the values relevant for any given setting can be found. I understand how this approach will cover the space of possible settings, though also think it may be less intuitive for readers as compared to an approach that produces results for specific named countries. However, the relevant question (as I see it) is whether the approach taken by the authors will give valid results, rather than how it compares to an alternative study design.

On this point I think the approach is generally valid. I am a bit worried about the text that discusses the overall average results, as these values may not be representative of any single setting. On this point, attention given to the stratified results (Table 5, in which results are reported according to different levels of the setting variables) is important. I would suggest that the authors review the text to make sure these nuances are discussed. It may also be useful to note that the variables used to describe different settings may not capture all relevant features of a given setting, and for this reason country-specific analyses could have value as follow-up studies.

ISSUE #4: Scaling of results.

The authors present their results per unit population (in this case, per 10 million of population). This makes sense to me.

(Remarks on code availability)

Reviewer #4 (Remarks to the Author):

Manuscript: Enabling higher levels of HIV viral suppression in Africa: A modelling study of the potential impact and cost-effectiveness of introduction of long-acting injectable lenacapavir plus cabotegravir (NCOMMS-25-35509-T)

Review date: 5/2025

SUMMARY

This manuscript describes a modeling analysis of the potential health and economic implications of using long-acting ARV agents (in place of current regimens) as part of routine ART in Sub-Saharan African countries. I have focused my review on a set of outstanding issues that remain from earlier reviews. These four points are referred to below, in the order presented in the most recent response to reviewers comments.

We thank the reviewer for their comments, all of which we find helpful.

ISSUE #1: Relevance of cost-effectiveness results estimated over a 10-year analytic horizon.

It is conventional for cost-effectiveness analyses to estimate results over a future period that is longer enough such that all (or at least the large majority) of health and economic consequences of the intervention are represented in the analysis. For most health interventions a lifetime horizon is the default option. Where an intervention impacts disease transmission this complicates matters, but the 50-year horizon adopted by this analysis appears sufficient to represent the large majority of consequences (and discounting means that differences in outcomes beyond 50 years will have a minimal impact). I do not agree with the position that 10-year outcomes will be more salient for policy-makers (with the implication that later outcomes should be ignored). There are a large number of health interventions that are widely accepted to be high value and that require up-front spending to produce health gains that are realized over the remainder of the individual's lifetime. I worry that additionally presenting the 10-year CEA results could be misleading for readers who may not appreciate the distinctions.

That said, one thing that would be useful to communicate to readers is the relative timing of costs and health benefits. This is partly achieved by Figure 1. I would suggest the authors review the results/discussion text to check whether there are edits that could be made to communicate these dynamics clearly.

Thanks – we have added a comment on the relative timing of costs and health benefits in the Discussion: “We find an immediate impact on HIV deaths with a relatively small short term budget impact of a 3.5% rise.”

ISSUE #2: Description of the study population.

I agree with the reviewer that the age of modelled individuals needs to be stated clearly in the main text. In the methods section I see that the authors refer to individuals “age 15 or above”. If this is correct then I think that is sufficient. The text that has been added which refers to individuals of “age -72” (line 647) worries me a bit, I feel this will confuse readers rather than clarify, and suggest omitting.

Elsewhere in the manuscript reference is made to individuals aged 15-49. I understand this to be due

to the need to compare model estimates to external estimates that adopt this age range, and think this is reasonable.

We have taken out the line about starting at age -72. Yes that is correct that we refer to age ranges of 15-49 in order to compare estimates with data sources.

ISSUE #3: Which (if any) countries does the analysis represent.

The point raised by the reviewer questions whether the model is a sufficient representation of individual countries. This is an important point, as the policy question addressed by the study has to be relevant to individual countries to be useful. The authors took the approach of defining a small number of model inputs that represent differences between settings, creating ranges for these variables (representing extreme low and high values for what might be expected in any country or sub-national setting), and then randomly sampling from these ranges to create a bank of results in which the values relevant for any given setting can be found. I understand how this approach will cover the space of possible settings, though also think it may be less intuitive for readers as compared to an approach that produces results for specific named countries. However, the relevant question (as I see it) is whether the approach taken by the authors will give valid results, rather than how it compares to an alternative study design.

On this point I think the approach is generally valid. I am a bit worried about the text that discusses the overall average results, as these values may not be representative of any single setting. On this point, attention given to the stratified results (Table 5, in which results are reported according to different levels of the setting variables) is important. I would suggest that the authors review the text to make sure these nuances are discussed. It may also be useful to note that the variables used to describe different settings may not capture all relevant features of a given setting, and for this reason country-specific analyses could have value as follow-up studies.

We say in the limitations section of the manuscript “Lastly, while there are benefits of modelling a range of setting scenarios representing the diversity of settings in ECSWA in that we can assess what attributes of a setting influence impact and cost-effectiveness, we recognise that we need to take care in interpretation of outcomes for an “average” setting scenario and that there would be some additional value in modelling the effect of lenacapavir + cabotegravir implementation in the context of a model calibrated to data from one or more specific countries in aggregate.”

ISSUE #4: Scaling of results.

The authors present their results per unit population (in this case, per 10 million of population). This makes sense to me.

Thanks